# Scalably Using Node Attributes and Graph Structure for Node Classification [note 1]

**DOI:** 10.3390/e24070906

**Published:** 2022-06-30

**Authors:** Arpit Merchant, Ananth Mahadevan, Michael Mathioudakis

**Affiliations:** Department of Computer Science, University of Helsinki, 00014 Helsinki, Finland; ananth.mahadevan@helsinki.fi (A.M.); michael.mathioudakis@helsinki.fi (M.M.)

**Keywords:** node classification, graph embedding, representation learning

## Abstract

The task of node classification concerns a network where nodes are associated with labels, but labels are known only for some of the nodes. The task consists of inferring the unknown labels given the known node labels, the structure of the network, and other known node attributes. Common node classification approaches are based on the assumption that adjacent nodes have similar attributes and, therefore, that a node’s label can be predicted from the labels of its neighbors. While such an assumption is often valid (e.g., for political affiliation in social networks), it may not hold in some cases. In fact, nodes that share the same label may be adjacent but differ in their attributes, or may not be adjacent but have similar attributes. In this work, we present JANE (Jointly using Attributes and Node Embeddings), a novel and principled approach to node classification that flexibly adapts to a range of settings wherein unknown labels may be predicted from known labels of adjacent nodes in the network, other node attributes, or both. Our experiments on synthetic data highlight the limitations of benchmark algorithms and the versatility of JANE. Further, our experiments on seven real datasets of sizes ranging from 2.5K to 1.5M nodes and edge homophily ranging from 0.86 to 0.29 show that JANE scales well to large networks while also demonstrating an up to 20% improvement in accuracy compared to strong baseline algorithms.

## 1. Introduction

Node classification involves an attributed network with a known graph structure, where each node is associated with a *label* (or *class*, a categorical variable), as well as other attributes. Moreover, labels are known only for some of the nodes and are considered unknown for others. Given the graph structure, the labels that are known for some of the nodes, and other attributes that are known for all nodes, the task is to predict the labels that are unknown. This task finds application in a wide range of domains, such as information networks [1], complex systems [2], protein function identification [3], medical term semantic classification [4] and disease prediction [5].

### 1.1. Previous Work

A common assumption in the literature is that adjacent nodes tend to have similar labels, often represented in theories of homophily [6] and social influence [7]. For instance, in a social network, the assumption stipulates that friends, i.e., close social connections, are likely to vote for the same political party as seen in Figure 1(left). Approaches that rely on this assumption typically consider node proximity and assign the same label to nearby nodes. For example, in label propagation, labels diffuse in an iterative fashion from labeled nodes to their unlabeled neighbors until convergence [8]. Other approaches induce label uniformity within cuts or clusters of the graph [9,10,11], or consider node proximity in a latent space that preserves graph distances, as in DeepWalk [12] and similar matrix factorization approaches [13].

However, the aforementioned methods ignore other node attributes, which can be detrimental. For example, Hamilton et al. [3] show that, for certain predictive tasks on citation and social graphs, a linear classifier that is built only on node attributes outperforms approaches, such as DeepWalk, which are based on node proximity but ignore node attributes. Moreover, although homophily is often observed in some classification tasks, it is not uncommon to find that adjacent nodes do not share a particular label and that, in such cases, other node attributes can serve as better label predictors than graph structure [14,15]. For example, two individuals may be friends (i.e., be connected on a social network) but vote for different political parties (‘label’)—something that could be better predicted by rich, node-level attribute data (e.g., geographic location, income, or profession). Therefore, for node classification, it is important to appropriately leverage both the proximity of nodes on the graph structure and other attributes.

Partially addressing this limitation, AANE [16] and DANE [17] combine low-dimensional encodings of node attributes with graph-distance-preserving node embeddings, and use them as input features for label prediction. However, they do not account for known labels during training, and thus potentially ignore information that would be useful in predicting the unknown labels. LANE [18] overcomes this limitation by learning the joint latent representations of node attributes, proximity, and labels. However, LANE does not directly address the node classification task, i.e., it does not optimize the conditional probability distribution of node labels given the node attributes and graph structure, but rather targets their joint distribution of all quantities.

In a different line of work, graph convolutional networks use graph topology for low-pass filtering on node features [19]. GAT [20] introduces an attention mechanism to learn weights and aggregate features. GraphSAGE [3] uses mean/max pooling to sample and aggregate features from nodes’ local neighbourhoods. However, these convolutions are equivalent to a repeated smoothing over the node attributes, and performance quickly degrades [21]. Subsequent approaches such as DiffPool [22] have sought to address this limitation, but these too aggressively enforce homophily and require that nodes with the same labels have similar graph and attribute representations. Recent work by AM-GCN [23] attempts to weaken this assumption by analyzing the fusing abilities of convolutional models. They define two modules, one each for the topology space and feature space, and adaptively combine them using an attention mechanism. We direct the reader to the survey paper of Xiao et al. [24]’s for a comparison of recent works on graph convolutions for the task of node classification.

### 1.2. Our Contribution

In this work, we develop an approach to node classification that can flexibly adapt and perform consistently well in a range of settings, from cases where node labels exhibit strong homophily (i.e., a node’s label can be determined by the labels of its neighbors) to cases where labels are independent of graph structure and solely determined by other node attributes, or vice versa, as well as cases that lie between these extremes. We propose a novel and principled approach to node classification based on a generative probabilistic model that jointly captures the role of graph structure and node similarity in predicting labels. Our analysis leads to JANE, a training algorithm that learns two unknown model parameters, namely, a latent node embedding U and weight matrices W of a neural network for the task of node classification. JANE combines node attributes with the embedding and iteratively updates both model parameters U and W with label information during training. We show that this flexibly adapts to the prediction task in a variety of cases, depending on the correlation between attributes and graph structure on node labels. Unlike the aforementioned approaches, which are heavily based on label homophily (e.g., label propagation), we account not only for the graph structure but also for node attributes, and flexibly and appropriately weigh each of them depending on the case. Moreover, unlike AANE [16] and DANE [17], our approach learns a low-dimensional node representation that is informed by labels, which is then used for prediction. Unlike LANE [18], we directly optimize the conditional probability of labels given the graph structure and node attributes, without necessarily enforcing homophily.

We summarize our main contributions below:
We define a generative model that formally captures a diverse set of relationships between graph structure, node attributes, and node labels.We describe a general algorithm to compute a good initial estimate of U and a training algorithm called JANE for node classification. We also design batching and minibatching variants of JANE that scale well to large graphs.From our experimental results, we present three findings. First, we demonstrate the shortcomings of existing approaches and versatility of JANE on synthetic datasets. Second, we empirically validate the performance of four variants of JANE on seven real-world datasets and compare to five standard baselines. Thirdly, we conduct an extensive empirical analysis characterizing the usefulness of a good initial node embedding, the importance of updates to the embedding during training, and the trade-off between preserving adjacency and label information on classification accuracy.


## 2. Problem Setting

Let us consider an undirected and connected graph $G=\left(V,E\right)$ of node size |V|=n. Let its structure be represented by the adjacency matrix $A=\left[a_{ij}\right]\in R^{n\times n}$. Denote $D=\mathrm{diag}\left(d_{1},d_{2},\ldots ,d_{n}\right)$ to be the degree matrix where $d_{i}=\sum_{j}a_{ij}$; and L=D−A as its unnormalized Laplacian matrix. Let $\lambda_{i}$ be the *i*-th smallest eigenvalue of L and $e_{i}$ its corresponding eigenvector.

Each node in the graph is associated with the following: *d* observed attributes $x_{}\in R^{d}$, *k* latent/unobserved node embeddings **u** $\in R^{k}$, and possibly an unobserved categorical variable $y_{}\in {\left\{0,1\right\}}^{M}$ (one-hot encoding) as the label from label-set $M=\left\{1,2,\ldots ,M\right\}$. For example, in citation graphs, with nodes corresponding to articles and edges to citations between articles, $x_{}$ capture observed quantities such as the bag-of-words representation of the article text, and label $y_{}$ denotes the research area of the article (e.g., ‘data mining’ or ‘machine learning’). The latent embedding **u** corresponds to the properties of the articles that are not directly captured by attributes $x_{}$ or label $y_{}$, but that could play a role in determining which articles are connected with a citation (as captured by adjacency matrix A) and to which research area $y_{}$ an article is deemed to belong. In terms of notation, to refer to the attributes of all nodes, we write $X=\left\{x_{i}\in R^{d},i\in \left\{1,\ldots ,n\right\}\right\}$ to denote the observed node attributes, $U=\left\{u_{i}\in R^{k},i\in \left\{1,\ldots ,n\right\}\right\}$ for the latent node embedding, and $Y=\left\{y_{i}\in {\left\{0,1\right\}}^{M},i\in \left\{1,\ldots ,n\right\}\right\}$ for the node labels. This naturally extends to the multi-label classification setting.

Having defined all elements in our setting, we now define the task that we address as Problem 1.

**Problem** **1 (Node-Classification).**
*Given adjacency matrix A, node features X, and labels YL for a subset L⊆V of nodes, predict labels YV\L for the remaining nodes V\L in the graph.*


## 3. Our Approach

Our approach for Problem 1 is based on a probabilistic generative model (described in Section 3.1) and its analysis (Section 3.2).

### 3.1. Model

Figure 2 pictorially illustrates our generative model. First, the adjacency matrix A of the graph is generated from the latent embedding U. Specifically, the probability that there is an edge between two nodes *i* and *j* is given by the inverse exponent of the squares l2-distance between their latent attributes **u** scaled by a factor $s^{2}$.(1)$$\mathrm{Pr}\left[\;\left(i,j\right)\in E\;|\;u_{i},u_{j};s\;\right]=p_{ij}=e^{\frac{-{∥u_{i}-u_{j}∥}^{2}}{s^{2}}}$$

This equips our model with the desirable property, common in many types of graph embeddings, that the closer the two nodes are in the Euclidean space of U, the higher the likelihood that they are connected in the graph and vice versa. Therefore, U represents a low-dimensional Euclidean embedding of the graph that preserves connectivity in the form of Equation (Equation 1). Moreover, since the existence of an edge is independent across pairs of nodes in this model, we have(2)$$\mathrm{Pr}\left[A|U;s\right]=\prod_{\left(i,j\right)\in E}p_{ij}\times \prod_{\left(i,j\right)\notin E}\left(1-p_{ij}\right).$$
Notice that, with regard to Equations (Equation 1) and (Equation 2), the scaling factor *s* can be absorbed into the embedding parameter U, i.e., any pair (U, *s*) of parameters is equivalent to the pair of parameters (U/*s*, 1). For this reason, we will omit *s* from the probability expressions that follow.

There are several other graph-generation models, such as the ϵ-neighbourhoods model, wherein nodes i,j are connected by an edge if ${∥u_{i}-u_{j}∥}^{2}\leq \epsilon$ [26]. However, the ϵ-neighbourhoods model often leads to several connected components. Moreover, Equation (Equation 2) is in line with typical assumptions in spectral graph theory [27].

Second, for fixed scale *s*, node labels Y are generated from X and U. This assumption provides two benefits: (i) it allows for labels to be determined by node attributes X (directly) as well as graph structure A (indirectly, via U); (ii) it allows for us to directly express and train the function of the conditional distribution $\mathrm{Pr}\left[Y|X,U\right]$, which we then employ for *node classification*, i.e., to predict unobserved node labels.

In this work, we assume that this conditional probability is given by a simple two-layer neural network,(3)$$\mathrm{Pr}\left[Y|X,U,W\right]=\sigma \left(\mathrm{ReLU}\left(\mathrm{CONCAT}\left[X,\;U\right]W^{\left(0\right)}\right)W^{\left(1\right)}\right)$$where σ denotes the softmax function and weight matrices $W=\{W^{\left(0\right)},W^{\left(1\right)}\}$ are parameters that control the effect of X and U on labels Y.

The reason for this choice is that we found this model to be sufficiently expressive for our empirical evaluation. We note that this conditional probability (RHS in Equation (Equation 3)) can be replaced with other complex models (e.g., neural networks with more hidden layers).

### 3.2. Algorithms

Given data $D=(X,A,Y_{L})$ as input, Problem 1 asks for predictions for $Y_{V\backslash L}$. Here, the latent embedding U and the weights of the neural network W, are considered as unknown parameters ⊆ = (U, W) of the model. Our approach, JANE, proceeds in two stages: first, a training stage, from which we learn the maximum likelihood estimates $\hat{\theta }=\left(\hat{U},\hat{W}\right)$ of the model (cf. Equation (Equation 3)); second, a prediction stage, in which we use $\hat{\theta }$ to predict the missing labels.

#### 3.2.1. Training

**Training Objective.** From the product rule of probability, the likelihood of the data D given parameters ⊆ is as follows:(4)$$\begin{array}{cc} \begin{array}{c} \mathrm{Pr}\left[D|\subseteq \right] \end{array} & =\mathrm{Pr}\left[X,A,Y_{L}|U,W\right]\\ \; & \propto \mathrm{Pr}\left[Y_{L}|X;U,W\right]\times \mathrm{Pr}\left[A|U\right] \end{array}$$

Taking the negative logarithm on both sides, we define the total loss as the sum of the loss with respect to node labels ($L_{\mathrm{label}}$) and the loss with respect to the graph structure ($L_{\mathrm{graph}}$).(5)$$\begin{array}{cc} \begin{array}{c} L_{\mathrm{total}} \end{array}\; & =-\log\mathrm{Pr}\left[D|\subseteq \right]\\ \; & =\underset{L_{\mathrm{label}}}{\underset{︸}{-\log\mathrm{Pr}\left[Y_{L}|X;U,W\right]}}\underset{L_{\mathrm{graph}}}{\underset{︸}{-\log\mathrm{Pr}\left[A|U\right]}} \end{array}$$

The goal is to identify parameter estimates $\hat{\theta }$ that minimize the objective function $L_{\mathrm{total}}$. We begin by describing a general routine to construct a good initial estimate of U (Algorithm 1). Then, we describe a training procedure to update $\hat{U}$ and $\hat{W}$ to their maximum-likelihood estimates of the model using stochastic gradient descent, starting from ${\hat{U}}_{\mathrm{initial}}$ (Algorithm 2).
**Algorithm 1:** INITIALIZE.**1 Input**: Adjacency Matrix A; Embedding method EMBEDDING∈ {Laplacian, Gosh}, embedding dimensionality *k*;**2 Output**: Latent attributes ${\hat{U}}_{\mathrm{initial}}$, Scale *s*;**3** s←1**4** $\bar{U}\leftarrow \mathrm{EMBEDDING}\left(\left(\frac{1}{s}\cdot A\right),\;k\right)$**5** $s\leftarrow \mathrm{MINIMIZE}\left(\mathrm{objective}=L_{\mathrm{edge}}+L_{\mathrm{nonedge}},\mathrm{constants}=A,\bar{U}\right)$// ((cf. Equation (Equation 7))**6** ${\hat{U}}_{\mathrm{initial}}\leftarrow \frac{1}{s}\cdot \bar{U}$, $s_{\mathrm{initial}}\leftarrow 1$// Rescale    **return**${\hat{U}}_{\mathrm{initial}},\;s_{\mathrm{initial}}$
**Algorithm 2:** JANE-U (Batching).
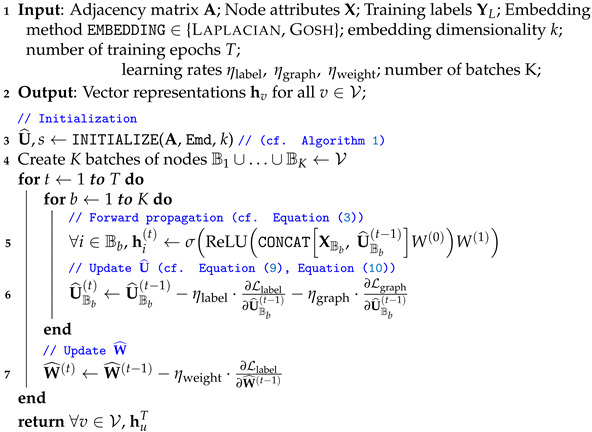


**Choosing $U_{\mathrm{initial}}$.** We construct an initial estimate of $\hat{U}$ and *s* in an unsupervised manner using only the adjacency information and (temporarily) ignore node labels. That is,(6)$$\begin{array}{c} {\hat{U}}_{\mathrm{initial}},s_{\mathrm{initial}}=\underset{U,s}{arg min}\;L_{\mathrm{graph}} \end{array}$$

Plugging in Equation (Equation 2), we get(7)$$\begin{array}{cc} \begin{array}{c} {\hat{U}}_{\mathrm{initial}},s_{\mathrm{initial}} \end{array} & =\underset{U,s}{arg min}\sum_{\left(i,j\right)\in E}\frac{{∥u_{i}-u_{j}∥}^{2}}{s^{2}}-\log\left(\prod_{\left(i,j\right)\notin E}\left(1-e^{\frac{-{∥u_{i}-u_{j}∥}^{2}}{s^{2}}}\right)\right)\\ \; & \leq \underset{U,s}{arg min}\underset{L_{\mathrm{edge}}}{\underset{︸}{\sum_{\left(i,j\right)\in E}\frac{{∥u_{i}-u_{j}∥}^{2}}{s^{2}}}}+\underset{L_{\mathrm{nonedge}}}{\underset{︸}{\sum_{\left(i,j\right)\notin E}e^{\frac{-{∥u_{i}-u_{j}∥}^{2}}{s^{2}}}}} \end{array}$$where the last inequality holds because $-\left(\log\left(1-x\right)\right)\leq x,x\in \left[0,1\right)$. The first term, $L_{\mathrm{edge}}$, captures the likelihood of observing all edges in the graph, while the second term, $L_{\mathrm{nonedge}}$, captures the likelihood of all non-edges (i.e., node pairs without an edge between them).

We obtain the initial estimates for U and *s* by iteratively minimizing the upper bound of Equation (Equation 7). In each iteration, we first fix the scale *s* and minimize over U to obtain $\bar{U}$; then, we fix $\bar{U}$ and minimize over *s* to obtain ${\hat{U}}_{\mathrm{initial}}$. We have explored two approaches to obtain $\bar{U}$, namely via *Laplacian eigenvectors* and *GOSH embeddings*, and we describe them below in this section. Moreover, we use Scipy’s implementation of the standard *Brent* bounded scalar optimization algorithm (https://docs.scipy.org/doc/scipy/reference/generated/scipy.optimize.minimize_scalar.html (25 May 2022)) to obtain *s* within each iteration. Note that, in practice, we observed that a single iteration is sufficient to obtain good initial estimates for U and *s* in both aforementioned approaches, and we thus use a single iteration for the initialization in all that follows. The initialization procedure is shown in Algorithm 1. Note that, for conventional simplicity, we always absorb the scale factor *s* into the initial embedding U at the end of the initialization procedure (Algorithm 1, Step 6).

We now proceed to describe the two approaches we have explored to obtain $\bar{U}$.

1. *Laplacian Eigenvectors* [25]: Observe that for an appropriate choice of scale, i.e., for smaller values of *s*, $L_{\mathrm{nonedge}}$ tends to 0 and the value of $L_{\mathrm{edge}}$ dominates Equation (Equation 7). Moreover, given that L is symmetric and $d_{i}=\sum_{j}a_{ij},\forall i\in \left[n\right]$, based on the results of Belkin et al. [26],$$\begin{array}{cc} \min_{\begin{array}{c} U:\forall l\in [k],\;\\ ∥u_{l}∥=1,\\ \sum_{p}\;u_{lp}=0 \end{array}}\;\sum_{\left(i,j\right)\in E}{∥u_{i}-u_{j}∥}^{2}\; & =\mathrm{tr}\left(U^{T}LU\right)=\sum_{l=1}^{k}\lambda_{l}. \end{array}$$
The condition $U:\forall l\in \left[k\right],\;\;∥u_{l}∥=1,\;\sum_{p}\;u_{lp}=0$ normalizes the columns of U, removes translational invariance, and centers the solution around ⊬. This result implies that the minimum value of $L_{\mathrm{edge}}$ in Equation (Equation 7) is the sum of the *k* smallest eigenvalues of the graph Laplacian. Moreover, this minimum value is achieved when columns of U are the corresponding eigenvectors. That is:(8)$$\bar{U}=\left[e_{1},e_{2},\ldots ,e_{k}\right].$$where $e_{i}$ is the *i*-th smallest spectral eigenvector. We make use of the above to obtain a heuristic value for $\bar{U}$. Specifically, in Step 4 of Algorithm 1, we set $\bar{U}$ to the Laplacian eigenvectors, effectively choosing to minimize only $L_{\mathrm{edge}}$ and ignore $L_{\mathrm{nonedge}}$. Note that when selecting the scale factor *s* at Step 5 we make use of the sum $L_{\mathrm{edge}}+L_{\mathrm{nonedge}}$, i.e., the full Equation (Equation 7).

2. *GOSH embeddings* [28]: GOSH is an efficient and state-of-the-art method to obtain embeddings that preserve vertex-to-vertex similarity measures. For our purposes, we use the normalized adjacency matrix of the graph as the similarity measure. While the objective function that is optimized by GOSH out-of-the-box is not identical to that of Equation (Equation 7), the resulting optimization performed by GOSH leads to embeddings that preserve both edge and non-edge information, in line with our objective in Equation (Equation 7). To further ensure that GOSH provides an embedding with a good objective value, we perform a grid-search over its hyperparameters and maintain the GOSH embedding with maximum objective value as per Equation (Equation 7). We have found this approach to work extremely well in practice.

*GOSH vs. Laplacian Embeddings:* We found experimentally that performing the initialization of U with GOSH rather than Laplacian embeddings consistently leads to a better performance. Therefore, for ease of presentation in what follows, we will use GOSH for the initialization procedure (Algorithm 1), unless explicitly mentioned otherwise. Beyond Laplacian and GOSH embeddings, this framework is flexible enough to support other unsupervised node embeddings, such as VERSE [29], Force2vec [30], which aim to preserve edge and non-edge information.

**Updating Model Parameters.** Next, we describe the training procedure for JANE to minimize the overall loss $L_{\mathrm{total}}$. Training begins with the values $U=U_{initial}$ and s=1 provided by Algorithm 1, with the scale *s* remaining fixed during training. During each training epoch, JANE starts by fixing $\hat{W}$ and updating its estimate $\hat{U}$ using gradient descent on $L_{\mathrm{total}}$ (cf. Equation (Equation 5)). This involves computing two gradients. First, the gradient of $L_{\mathrm{label}}$ w.r.t. the *i*-th row of the current latent embedding, ${\hat{u}}_{i}$, is obtained using backpropagation as follows,(9)$$\frac{\partial L_{\mathrm{label}}}{\partial {\hat{u}}_{i}}=\sum_{l\in |Y_{L}|}\left(\sum_{r=1}^{M}\left(a_{lr}^{\left(1\right)}-Y_{lr}\right)W_{ri}^{\left(1\right)}\right)\times \left(\frac{a_{li}^{\left(0\right)}}{1+\exp\left(-z_{li}^{\left(0\right)}\right)}\right)\times \left(\sum_{p=1}^{d+k}\left[W_{il}^{\left(0\right)}\right]\right)$$where *a*^(0)^, *a*^(1)^ are activations from the hidden and output layers, and $z_{}^{\left(0\right)}$ is the weighted sum from the input layer. The index *i* ranges over all nodes with known labels, and *r* indexes over the *M* different classes available for prediction. Second, the gradient of (upper-bounded) $L_{\mathrm{graph}}$ in Equation (Equation 7) w.r.t. the *i*-th row of the current latent embedding, ${\hat{u}}_{i}$, is as follows.(10)$$\frac{\partial L_{\mathrm{graph}}}{\partial {\hat{u}}_{i}}=\left(\sum_{j:\left(i,j\right)\in E}2\times \frac{{\hat{u}}_{i}-{\hat{u}}_{j}}{s^{2}}\right)+\left(\sum_{j:\left(i,j\right)\notin E}-2\times \frac{{\hat{u}}_{i}-{\hat{u}}_{j}}{s^{2}}\times e^{-\frac{{∥{\hat{u}}_{i}-{\hat{u}}_{j}∥}^{2}}{s^{2}}}\right)$$where indices *i* and *j* range over all nodes in V. The learning rates corresponding to the gradients above are denoted with $\eta_{\mathrm{label}},\eta_{\mathrm{graph}}$.

Once $\hat{U}$ is updated, JANE treats it as fixed and updates the weights $\hat{W}$ of the neural network using stochastic gradient descent for $L_{\mathrm{total}}$ (cf. Equation (Equation 5)). Since $L_{\mathrm{graph}}$ is independent of W, the optimization problem reduces to the following:(11)$$\begin{array}{c} \hat{W}=\underset{W}{arg min}L_{\mathrm{label}}=\underset{W}{arg min}-\log\left(\mathrm{Pr}\left[Y_{L}|X;\hat{U},W\right]\right) \end{array}$$
The gradient of $L_{\mathrm{label}}$ w.r.t $\hat{W}$ is computed using standard backpropagation (as in Step 7) with learning rate $\eta_{\mathrm{weight}}$.

**Batching.** To reduce memory overhead and improve scalability to larger graphs, we adapt our training algorithm to allow the forward and backward propagations described above to take place in *K* smaller batches of nodes, where $V=B_{1}\cup \ldots \cup B_{K}$. Algorithm 2 gives the pseudocode for JANE using this batched approach. This takes advantage of the observation that, during any current iteration of gradient descent update, ${\hat{U}}_{B_{b}}$ for a particular batch of nodes $b\in \left[K\right]$ does not depend on other nodes in the graph. An analogous minibatching variant called JANE-MU differs in only one aspect from JANE, namely that it updates $\hat{W}$ for every batch during each iteration. For completeness, we provide the pseudocode for the minibatching variant in Algorithm 3.

#### 3.2.2. Prediction

Given maximum-likelihood estimates ${\hat{U}}^{\left(T\right)}$ and ${\hat{W}}_{\left(T\right)}$ at the end of *T* epochs, we predict labels $\begin{array}{c} {\hat{Y}}_{V\backslash L} \end{array}$ for all nodes in $\begin{array}{c} V\backslash L \end{array}$ using the softmax function applied row-wise(12)$$\begin{array}{c} {\hat{Y}}_{V\backslash L}=\underset{r\in M}{arg max}\sigma {\left(\mathrm{ReLU}\left(\mathrm{CONCAT}\left[X_{V\backslash L},{\hat{U}}_{V\backslash L}^{\left(T\right)}\right]{\hat{W}}_{\left(T\right)}^{\left(0\right)}\right){\hat{W}}_{\left(T\right)}^{\left(1\right)}\right)}_{r}, \end{array}$$where $X_{V\backslash L}$ and ${\hat{U}}_{V\backslash L}^{\left(T\right)}$ are the corresponding node features and latent embeddings.

#### 3.2.3. Complexity Analysis

Computing ${\hat{U}}_{\mathrm{initial}}$ using the GOSH embedding has a time complexity of $O\left(|V|+|E|\right)$ [28]. We assume that the dimensionality of the hidden neural layers are of the same order as $O\left(d+k\right)$. A forward pass of JANE requires $O\left({\left(d+k\right)}^{2}L|V|+\left(d+k\right)\left|E\right|\right)$ where *L* is the number of layers, $O\left(\left(d+k\right)\left|E\right|\right)$ is the cost of the first layer’s linear mapping and $O\left({\left(d+k\right)}^{2}\left|V\right|\right)$ is the cost of each subsequent hidden layers.
**Algorithm 3:** JANE-U-M (Mini-Batching).
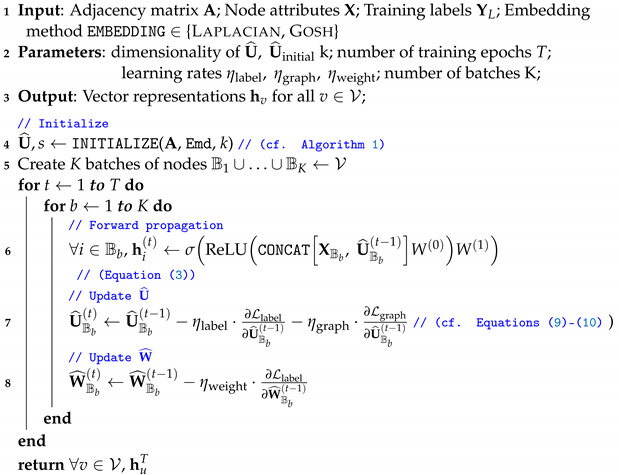



## 4. Experiments

In this section, we empirically evaluate the performance of JANE on synthetic and real-world datasets.

### 4.1. Algorithms

**Variants of** **JANE**. We conducted experiments for four variants of JANE, depending on the choice ${\hat{U}}_{\mathrm{initial}}$ and the training procedure.

JANE*-R:* This chooses a random matrix of appropriate dimensions (n×k) as ${\hat{U}}_{\mathrm{initial}}$ and trains according to Algorithm 2 in a batched manner.JANE*-U:* This computes a GOSH embedding [28] based on a hyperparameter search that minimizes Equation (Equation 7) as ${\hat{U}}_{\mathrm{initial}}$ and trains according to Algorithm 2 in a batched manner.JANE*-NU:* This computes a GOSH embedding [28] based on a hyperparameter search that minimizes Equation (Equation 7) as ${\hat{U}}_{\mathrm{initial}}$. However, ${\hat{U}}_{\mathrm{initial}}$ does not update during training (i.e. ignores Step 7). In effect, this becomes a simple feed-forward multi-layer perceptron.JANE*-U-M:* This is similar to JANE-U, except that it trains in a minibatched fashion (cf. Algorithm 3).

**Baselines.** We compare the performance of JANE with five baselines divided into three broad groups.


*Graph-structure agnostic algorithms:* Random Forest (RF) [18] trains a random forest on the attribute matrix X and does not incorporate adjacency information.*Node-attribute agnostic algorithms:* Label Propagation (LP) [8] chooses labels for nodes based on the community label of its r-hop neighbors. DeepWalk (DW) [12] encodes neighbourhood information via truncated random walks and uses a Random Forest Classifier for label prediction. These do not incorporate node attributes.*Graph-convolution based algorithms:* GCN [31] and GraphSAGE (mean aggregator) [3] convolve over node attributes using the adjacency matrix. We acknowledge that this is extremely active area of research today and there exist several advancements that demonstrate improved performance in terms of training efficiency [32,33] and low homophily datasets [34,35]. However, GCN and GraphSAGE continue to prove to be strong benchmarks, and so we empirically compare with them to demonstrate our central point: JANE is a simple algorithm that efficiently scales and is competitive with the state-of-the-art on a variety of datasets.


**Implementation.** We implemented Random Forest using Scikit-Learn [36]. All other baselines and all variants of JANE were implemented using Pytorch Geometric [37]. In all our experiments, as is standard, all approaches only received the adjacency matrix of the graph A and the node attributes X as input. We grid searched over the hyperparameter space to find the best setting for all our baselines.

**Hardware.** We performed all experiments on a Linux machine with 32 cores, 32GB RAM and a NVIDIA A100 GPU.

### 4.2. Node Classification on Synthetic Data

The goal of these experiments was two-fold—(1) demonstrate the fundamental differences between and limitations of existing classification approaches using synthetic datasets wherein labels derive from only X, only U, or partly from both; and (2) show the strengths and general-purpose nature of JANE vis-à-vis the source of node labels.

**Synthetic Datasets.** Figure 3 describes representative synthetic datasets generated according to the model described in Section 3.1. We set the number of individual node features |X|=d=2 and number of latent features |U|=k=2. We generated these X and U from gaussian distributions for n=200 points, each of which belonged to one of M=4 classes and set the scale $s^{2}=1$. The adjacency matrix was probabilistically created from U according to Equation (Equation 1). An influence parameter, $\alpha \in \left[0,1\right]$, controlled the degree to which node labels derive from X or U: α=0.0 signifies that they derive only from U and are independent of X; α=1.0 that they derive only from X and are independent of U; and α=0.5 that they derive equally from X and U (specifically, without loss of generality, only the first feature from X and U contributes to label assignment). Figure 3a depicts an instance of X and U each for $\alpha =\left\{0.0,0.5,1.0\right\}$, respectively. The colors of points represent classes. We constructed the adjacency matrix from U as per Equation (Equation 2). Figure 3b depicts an instance of the corresponding graphs.

**Implementation Details.** We used Scikit–Learn’s make_classification to generate these datasets. The approaches did not have access to α or U. JANE was trained as a two-layer neural network for a maximum of T=200 epochs with dropout of 0.2 for each layer, weight decay of 0.05, and learning rate of 0.005 using Adam. We set the number of eigenvectors as k=2 and chose a scaling factor of $s^{2}=0.01$.

**Performance.** Figure 3c shows performance as a function of increasing training set sizes.

α=0.0: LP and DW infer that labels derive from A (indirectly). GCN converges attribute values of nodes in the same cluster but is not perfectly accurate because X does not correlate with Y. LANE forces the proximity representation to be similar to the attribute representation, and then smoothens it using the labels. It does not perform well, since there is no correlation between them.α=0.5: LP, DW are able to correctly classify nodes belonging to 2 out of 4 classes, i.e. precisely those nodes whose labels are influenced by U. Conversely, LANE is able to classify those nodes belonging to two classes of nodes that correlate with X. GCN smoothens the attribute values of adjacent nodes, and thus can correctly infer labels correlated with X.α=1.0 LP and DW reduce to random classifiers since adjacent nodes do not have similar labels. GCN reduces to a nearly random classifier because, by forcing adjacent nodes with different attribute values to become similar, it destroys the correlation between X and the labels.

In each of the three cases, JANE-NU and JANE achieve perfect accuracy because they flexibly learn during training whether labels correlate or partially correlate with X, A (indirectly). While these datasets are simplistic in nature and represent hard cases of homophily and heterophily, it is interesting to find that a simple MLP such as JANE performs well where GCN and GraphSAGE do not. This demonstrates how the homophily assumption—requiring nodes with a similar proximity and attributes to have the same labels—limits the performance of other approaches.

### 4.3. Node Classification on Real-World Data

We seek to understand: (1) to what extent JANE can capture real graph structures and their correlations with node labels, and (2) how well JANE compares with our baselines on these datasets.

#### 4.3.1. Datasets

We evaluated seven real datasets of sizes ranging from 2.5 thousand to 1.5 million nodes. If the original graph was disconnected, we extracted its largest connected component along with the corresponding node attributes and labels (This resulted in different datasets and performances compared to the results reported in prior works [31,34,38]). Table 1 summarizes the dataset statistics.

*Citation Networks:* Cora, Citeseer, PubMed [1] represent academic papers as nodes, edges denote a citation between two nodes, node features are 0/1-valued sparse bag-of-words vectors and class labels denote the subfield of research to which the papers belong.*Social Networks:* Flickr denotes users of the social media site that post images as nodes, edges represent follower relationships, and features are specified by a list of tags reflecting the interests of the users [39]. The labels used to predict are pre-defined categories of images.*Squirrel:* This is a Wikipedia dataset [40] where nodes are web pages, edges are mutual links between pages, features correspond to informative nouns in the pages, and labels are categories based on the average number of monthly views of the page.*Yelp [41] and Amazon [38]:* The multi-label classification task in these datasets is to predict the types of business or product categories given customer or buyer reviews and friendship or interaction relationships, respectively.

#### 4.3.2. Experimental Setup

For the citation datasets, we used the same train–validation–test splits as in Yang, et al. [42], minus the nodes, which do not belong to the largest connected component. These comprise of 20 samples for each class and represent 5% of the entire dataset. We use 500 additional samples as a validation set for hyperparameter optimization as per Kipf, et al. [31] to enable fair comparison. For all other datasets, we use the training and validation splits reported in their original works. We evaluate the performance of all approaches on the remaining nodes of the graph. For each dataset, we set k=128 dimensions for $\hat{U}$. We perform a grid search over the hyperparameter space defined by hidden dimension in $\left\{128,256,512\right\}$, dropout in $\left\{0.0,0.1,0.2,0.3\right\}$, and learning rates $\left\{0.00001,0.0001,0.001,0.01\right\}$ with weight decay set to 0.0005.

#### 4.3.3. Performance Analysis

Table 2 reports the average test micro-F1 accuracy scores for each variant of JANE and the baselines. Values in bold denote the algorithm that performed best for each dataset. We make the following observations: (1) Choosing a good ${\hat{U}}_{\mathrm{initial}}$ is important for classification accuracy. Starting from a random matrix and updating it during training results in poorer performance compared to using the GOSH embedding with or without updates. For instance, JANE-R achieves 58.50% on Cora compared to JANE-U and JANE-NU, which achieve 77.78% and 77.75%, respectively. This trend was observed across datasets. (2) The unsupervised GOSH embedding, by design, preserves adjacency information. Thus, even without further updates to ${\hat{U}}_{\mathrm{initial}}$, it performed well in the classification task across datasets. In particular, we observe that it beats the other algorithms in the case of Citeseer with 66.58%. (3) However, updates to ${\hat{U}}_{\mathrm{initial}}$ during training can improve test accuracy for some datasets (that is, compared to not updating). This is particularly evident in PubMed where JANE-U obtains 77.56% compared to JANE-NU’s 76.70%. This is not just influenced by the properties of the dataset, but also the choice of ${\hat{U}}_{\mathrm{initial}}$ because when using eigenvectors, the performance gains for JANE-U are consistently and significantly higher than those for JANE-NU. (4) JANE-U-M significantly outperforms all other baselines on the larger datasets Yelp and Amazon, with 60.70% and 77.23% accuracy scores, respectively. This represents upto 18.05% and 7.66% improvement over the second best baseline. This is because JANE-U-M updates more frequently, i.e., for each batch in every epoch, and thus learns better. (5) We find that GCN beats other algorithms in the case of the low-homophily dataset Squirrel. JANE-U-M (42.65%) outperforms GraphSAGE (41.44%) on Squirrel, but is unable to correlate similarity of attributes of nodes 1-hop away with labels, which GCN, by way of convolutions, has an increased capture ability.

#### 4.3.4. Ablation Study on Choosing ${\hat{U}}_{\mathrm{initial}}$


As described in Section 3.2, in all results reported to date, our method of choice to obtain ${\hat{U}}_{\mathrm{initial}}$ was to invoke Algorithm 1 with GOSH, and after a grid-search of GOSH hyperparameters, chose the GOSH embedding that leads to the highest $L_{\mathrm{graph}}$ (Equation (Equation 7)). In this passage, we address the question of whether the entire $L_{\mathrm{graph}}$ expression is necessary to obtain a good initial embedding ${\hat{U}}_{\mathrm{initial}}$, or if it would suffice to use only the edge ($L_{\mathrm{edge}}$) or non-edge ($L_{\mathrm{nonedge}}$) information for the same purpose. In Figure 4, we report the test accuracy of JANE-U for three different ${\hat{U}}_{\mathrm{initial}}$, which aim to minimize $L_{\mathrm{edge}}$, $L_{\mathrm{nonedge}}$ and $L_{\mathrm{graph}}$, respectively (cf. Equation (Equation 7)). We computed this using a grid search over GOSH’s hyperparameters for each loss function. Then, we trained JANE-U as per Algorithm 2 over a grid of training hyperparameters and report best results. We find that, in the case of graphs with high homophily, such as Cora, Citeseer, and PubMed, preserving the overall adjacency information results in better accuracy compared to preserving only edge or non-edge information. On the other hand, for low-homophily graphs such as Squirrel, JANE-U performs similarly when using a ${\hat{U}}_{\mathrm{initial}}$ that minimizes any of the three loss functions with similar observations for Flickr.

#### 4.3.5. Ablation Study on Updating $\hat{U}$

Figure 5 shows the change in $L_{\mathrm{label}}$ and $L_{\mathrm{graph}}$ before and after training for 200 epochs. We used this to analyze how different variants of JANE learn the tradeoff between fitting to adjacency and label information. For instance, since JANE-NU does not update $\hat{U}$ during training, its $L_{\mathrm{graph}}$ remains constant, but $L_{\mathrm{label}}$ reduces as expected. For other variants, both $L_{\mathrm{graph}}$ and $L_{\mathrm{label}}$ reduce over time. JANE-R starts with a higher $L_{\mathrm{graph}}$ (e.g., 31.16 vs. 11.78 for other variants of JANE in Cora), but all models begin with the same $L_{\mathrm{label}}$ because the initial W matrix of the neural network is random. We find that, for example, in the case of Yelp, the GOSH embedding achieves a stable tradeoff between capturing adjacency and label information, because $L_{\mathrm{graph}}$ only marginally improves. However, it achieves the best test accuracy (60.70%, Table 2). On the other hand, JANE-U updates ${\hat{U}}_{\mathrm{initial}}$ significantly during training in order to reach a stable tradeoff and good test accuracy. Thus, we conclude that JANE flexibly adapts to various datasets during training.

#### 4.3.6. Relationship between $L_{\mathrm{graph}}$ and Accuracy

Figure 6 depicts how $L_{\mathrm{graph}}$, represented by a blue line, and validation accuracy, represented by a red line, change during the training phase of JANE. Each row corresponds to a specific algorithm (e.g., first row depicts JANE-R on PubMed and Amazon) and each column corresponds to a specific dataset (e.g., first column depicts JANE-R, JANE-U, and JANE-U-M on PubMed). We observe that JANE-R’s accuracy continues to improve while $L_{\mathrm{graph}}$ reduces over time, indicating that it is learning well. After 200 epochs, however, its micro-F1 score (51.42%) remains lower than that of JANE-U (77.56%) for PubMed. For Amazon, even though it starts with a larger $L_{\mathrm{graph}}$ (7304 vs. 3066), the updates to ${\hat{U}}_{\mathrm{initial}}$ provide it with sufficient information, such that its performance (63.39%) is comparable to JANE-U (69.53%), but not JANE-U-M (77.23%). JANE-U-M learns very quickly and achieves a good accuracy score within a relatively low number of epochs. Lastly, we observe, e.g., in PubMed, that JANE-U and JANE-U-M show diminishing returns in terms of accuracy after reaching a peak at 30 and 87 epochs, respectively, even though $L_{\mathrm{graph}}$ continues to reduce. This implies that fitting to adjacency is only useful up to a certain extent and indicates that an early stopping criterion based on this may be beneficial to computational efficiency.

#### 4.3.7. Runtime and Memory

Figure 7 plots the average time for a single training epoch for the various neural-network training algorithms on all datasets. This does not include the time to generate GOSH [28] embeddings, which range from 0.03 s for Cora to 17.6 s for Amazon over 200 epochs. JANE-NU is the fastest, since it is a vanilla feed-forward neural network and does not update ${\hat{U}}_{\mathrm{initial}}$. In practice, this results in a 20× to 333× depending on the size of the dataset. For the small-to-medium-sized datasets such as Cora up to Flickr, JANE-U and JANE-U-M are up to 7× and 2× faster than GCN and GraphSAGE and comparable or marginally slower on the two large datasets, Yelp and Amazon. However, it should be noted that both GCN and GraphSAGE use neighborhood sampling during the aggregation and inference steps, whereas JANE-U and JANE-U-M compute the full gradients for every batch in each epoch in a serial manner and do not employ a stopping criterion for updating $\hat{U}$, which would significantly speed up training. JANE is faster during inference in comparison to GCN and GraphSAGE. Thus, JANE is scalable to large graphs while providing strong performance. Moreover, the peak GPU memory utilization ranges from 54.2 MB to 68 GB for Cora and Amazon, respectively.

## 5. Conclusions

In this paper, we developed an approach to node classification that flexibly adapts to settings where labels are strongly correlated to graph structure, on one hand, and to graphs where labels are strongly correlated to node attributes, on the other. We propose a generative framework to demonstrate how graph structural information and node attributes both, can jointly influence node labels Even simple instances of such situations, as shown in Figure 1 and empirically evaluated in Figure 3b, severely affect the performance of standard baselines. Our principled approach, JANE, starts with an initial unsupervised GOSH embedding that captures adjacency information. Then, jointly with attributes, it updates the initial embedding to also incorporate label information for the node classification task, leading to a strong performance in a variety of datasets. Given its simplicity, scalability, and performance, JANE can serve as a competitive algorithm for node classification and as a useful starting point when designing models that holistically account for different sources of node labels and go beyond requiring or enforcing homophily.

There are two main directions for future work. The first is to incorporate node subsampling [38] or graph subsampling [43] in the computation of Equation (Equation 7) to further reduce the computational bottleneck of JANE-U and JANE-u-M and scale to very large graphs. The second direction is to implement a distributed version of JANE-U to enable scaling to multi-GPU clusters.

## Figures and Tables

**Figure 1 entropy-24-00906-f001:**
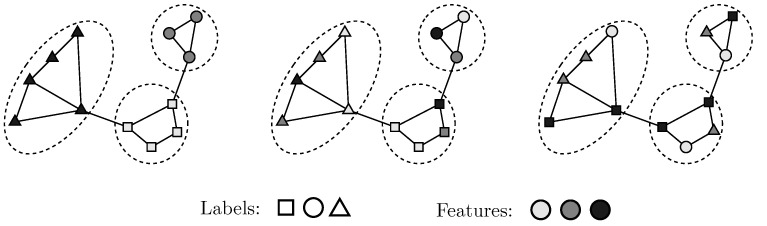
Nodes with the same labels may be adjacent and have similar features (**left**), or may be adjacent but have different features (**center**), or may not be adjacent but have similar features (**right**). Features are depicted by colors and labels are depicted by shapes. GCN and GraphSAGE can perform well in the first two cases but not the third because they predict node labels by aggregating the features of adjacent nodes.

**Figure 2 entropy-24-00906-f002:**
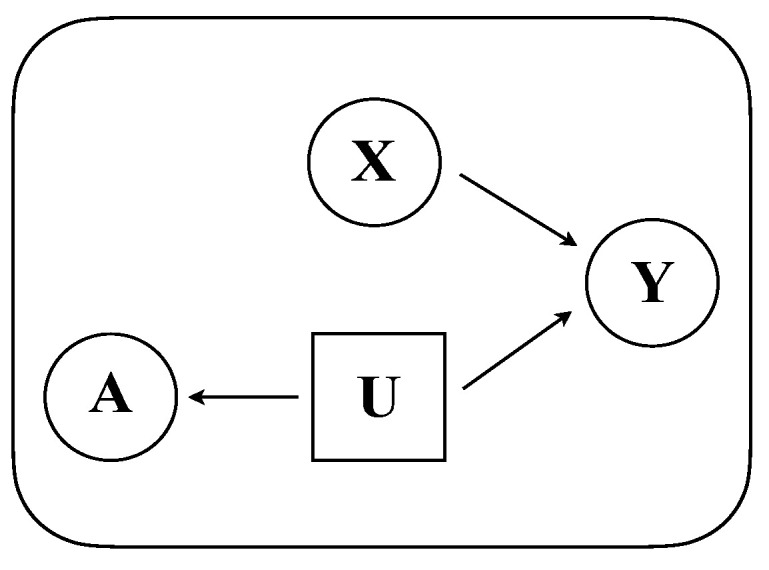
Visual illustration of the generative framework. Observed node attributes X (represented by circular box) and unobserved (latent) embeddings U (represented by square box) jointly generate node labels Y. The (observed) adjacency matrix A is generated from U and indirectly correlates with Y (via U). Reprinted with permission from Springer Nature Customer Service Centre GmbH: Springer, International Conference on Complex Networks and Their Applications, Joint Use of Node Attributes and Proximity for Node Classification, Merchant and Mathioudakis [25], 2021.

**Figure 3 entropy-24-00906-f003:**
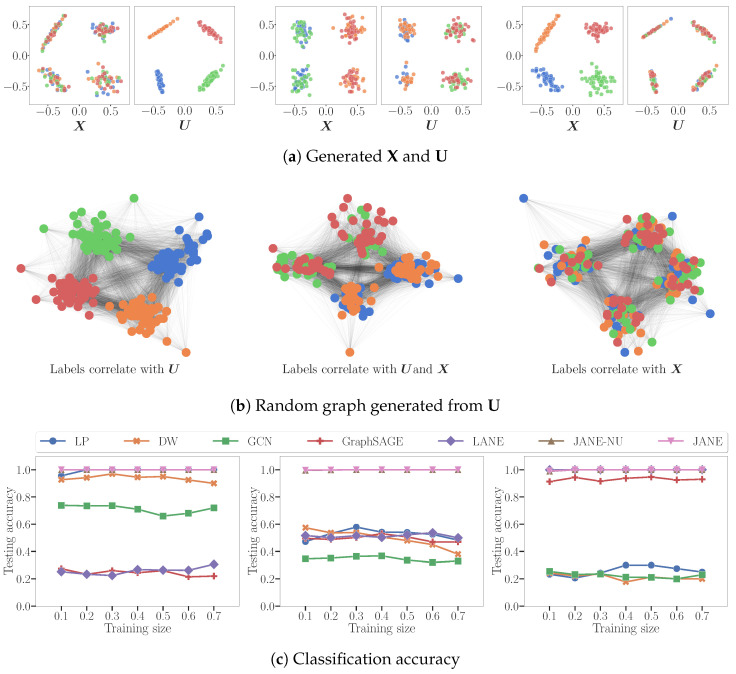
(**a**) depicts three synthetically generated datasets X and U with class labels Y influenced only by U (α=0.0, left), partly by U and partly by X (α=0.5, center), and only by X (α=1.0, right). (**b**) shows an instance of the corresponding graph generated from U according to Equation (Equation 2). (**c**) compares the node classification accuracy of JANE and JANE-NU, with the baselines averaged over five random train-test splits. Reprinted with permission from Springer Nature Customer Service Centre GmbH: Springer, International Conference on Complex Networks and Their Applications, Joint Use of Node Attributes and Proximity for Node Classification, Merchant and Mathioudakis [25], 2021.

**Figure 4 entropy-24-00906-f004:**
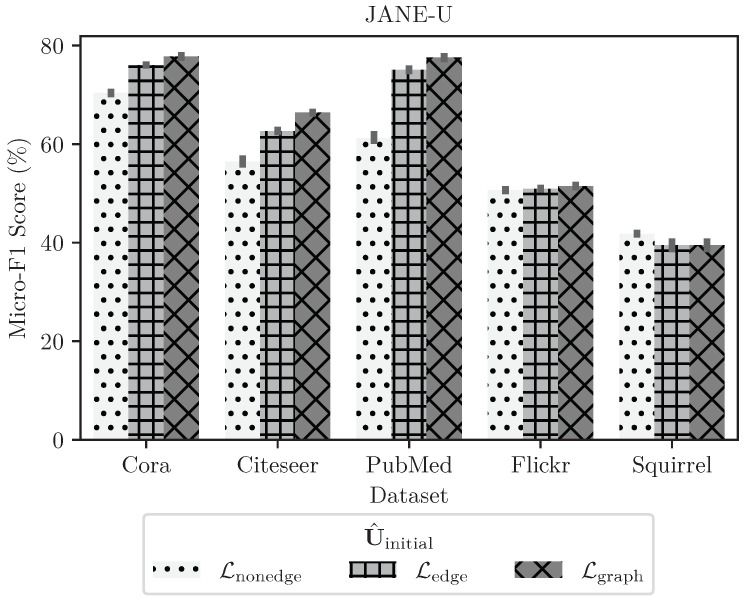
Test micro-F1 scores for JANE-U where ${\hat{U}}_{\mathrm{initial}}$ is that which minimizes $L_{\mathrm{edge}}$, $L_{\mathrm{nonedge}}$, or $L_{\mathrm{graph}}$. Jointly preserving edge as well as non-edge information results in best performance.

**Figure 5 entropy-24-00906-f005:**
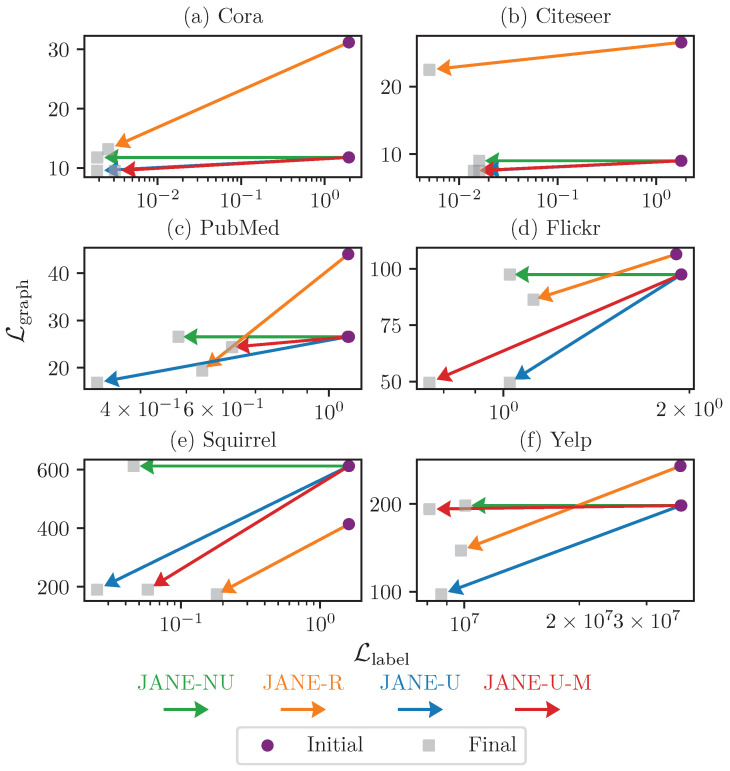
Change in $L_{\mathrm{label}}$ (X-axis) and $L_{\mathrm{graph}}$ (Y-axis) due to training for JANE-R, JANE-NU, JANE-U, and JANE-U-M. The purple circle denotes the loss value before the start of training and the grey square denotes the loss value after training for 200 epochs. JANE learns to fit to both the labels and adjacency during training.

**Figure 6 entropy-24-00906-f006:**
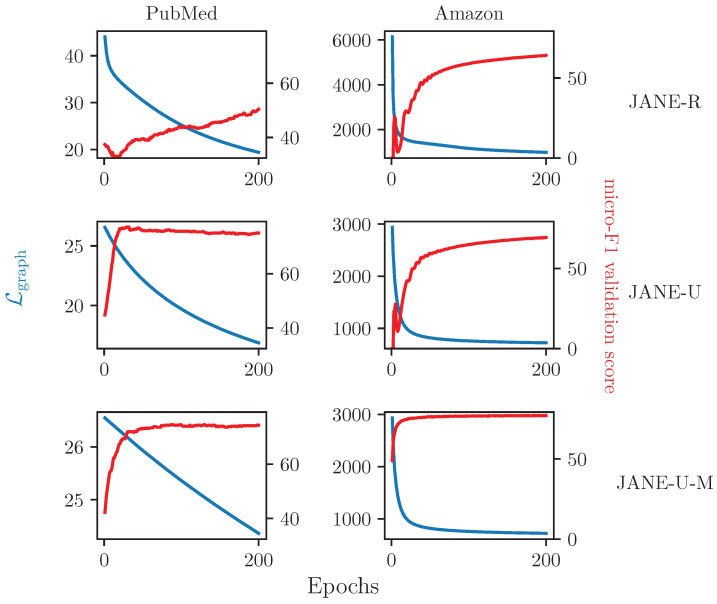
Decrease in $L_{\mathrm{graph}}$ (blue line, left Y-axis) and increase micro-F1 validation accuracy (red line, right Y-axis) as a function of training epochs (X-axis) for JANE-R, JANE-U, and JANE-U-M on Pubmed and Amazon datasets.

**Figure 7 entropy-24-00906-f007:**
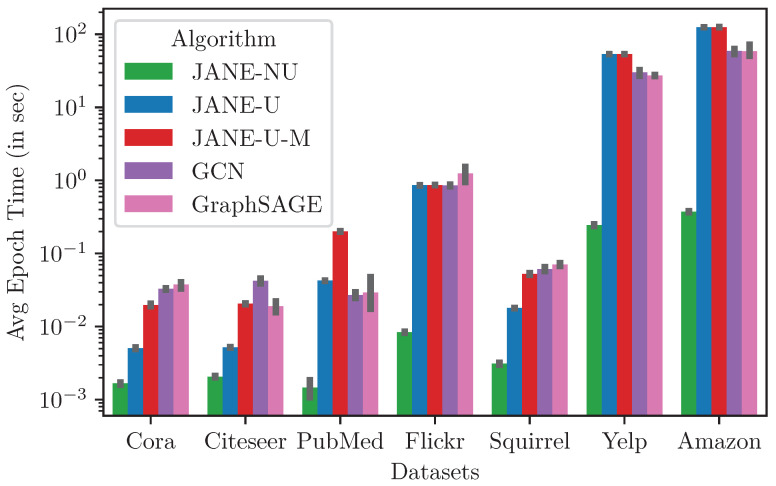
Average training time per epoch (in seconds) of training algorithms on different datasets.

**Table 1 entropy-24-00906-t001:** Dataset statistics: number of nodes |V|, number of edges |E|, number of classes, number of features, and edge homophily of the graph, respectively. Edge homophily is defined as the fraction of edges in a graph where both nodes have the same class label. We do not report homophily scores for the multi-label datasets Yelp and Amazon.

Graph	Size		Properties		
	|V|	|E|	Number of Classes	Number of Features	Edge Homophily
Cora	2485	5069	7	1428	0.86
Citeseer	2110	3668	6	3669	0.80
PubMed	19,717	44324	3	500	0.86
Squirrel	5201	401,907	5	2089	0.29
Flickr	89,250	989,006	7	500	0.41
Yelp	703,655	13,927,667	100	300	NA
Amazon	1,066,627	263,793,649	107	200	NA

**Table 2 entropy-24-00906-t002:** Comparison of test micro-F1 scores with baseline algorithms. Accuracy numbers in blue represent the best model for each dataset. Results are averaged over 10 runs.

Algorithm	Cora	Citeseer	Squirrel	PubMed	Flickr	Yelp	Amazon
RandomForest	56.19±0.2	49.20±0.6	33.31±0.5	73.70±0.8	45.80±0.0	42.55±0.1	59.12±0.1
LabelPropagation	70.53±0.0	64.30±0.0	32.37±0.0	70.50±0.0	46.96±0.0	0.90±0.0	0.02±0.0
DeepWalk	36.74±1.6	33.73±2.2	33.78±1.1	69.34±1.3	51.13±0.1	NA	NA
GCN	77.34±0.3	63.90±0.6	47.44±2.4	76.48±0.2	42.67±0.4	39.58±1.6	12.22±0.1
GraphSAGE	74.97±1.5	58.90±1.2	41.44±0.8	73.72±0.7	51.00±0.6	39.07±3.2	68.23±0.4
JANE-R	58.40±0.8	49.58±0.4	31.39±1.4	51.42±1.9	45.39±0.4	39.03±2.2	63.39±0.6
JANE-NU	77.75±0.3	66.58±0.2	42.32±0.5	76.70±0.7	51.44±0.2	40.98±3.4	69.57±0.2
JANE-U	77.78±0.3	66.40±0.1	41.81±0.2	77.56±0.3	51.49±0.2	38.76±1.2	69.53±0.2
JANE-U-M	78.10±0.1	66.30±0.2	42.65±1.2	77.14±0.6	50.67±0.2	60.70±0.6	77.23±0.1

## Data Availability

Real-world datasets that we use in our study are publicly available. Code for generating synthetic data and for our experiments is available at https://version.helsinki.fi/ads/jane (25 May 2022).

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
