# Peer review of "Scalably Using Node Attributes and Graph Structure for Node Classification†"

_entropy, 2022, doi:10.3390/e24070906_

Round 1

Reviewer 1 Report

This manuscript is devoted to the problem of classification of graph nodes. The authors propose a new approach to the node classification, which uses both information about class labels for some nodes and information about node attribute values.

It should be noted that this problem is an important and relevant task for many data analysis applications.

The advantages of the paper include its well structure and the clear statement of the research problem. In addition, the authors presented a quite complete overview of the study on the topic of node classification problem based on the graph structure and node attributes. The algorithm proposed by the authors uses a probabilistic generative model. The algorithm consists in finding the parameters of a two-layer neural network, which, over the learning process, minimize the objective function (5). The authors describe the process of choosing the initial values ​​of the parameters and the process of updating the parameters of the model using the backpropagation method. Moreover, to reduce the use of memory in the analysis of large graphs, a scheme for adapting the learning algorithm is presented. The analysis of the algorithm shows its polynomial complexity.

The authors conducted experiments on both synthetic and real datasets of various sizes, the largest of which has more than one million nodes and 250 million connections. Comparison with benchmark algorithms showed the advantage of the algorithm proposed by the authors. The analysis of the algorithm given in the paper is detailed, complete and includes also the time and capacity complexity analysis. Both the data and the program code are placed by the authors in an open repository.

I could not find any shortcomings in the work.

I believe that the manuscript corresponds to the topics of the journal, is of great interest to readers and can be published in its present form.

Author Response

We are grateful for your review. 

Reviewer 2 Report

1)Background and motivation should be enhanced significantly.

2)The contributions could be summarized in short. For example, some contributions can be combined together.

3) Could you please prove the hardness for node-classification.?  I suggest the relevant analysis to be added in this section.

4) The future work should be discussed in the conclusion section.

Author Response

Please see attachment below

Reviewer 3 Report

This paper proposed a method for node classification with a good article structure and experimental design. However, it still needs some improvement.

1.      In Introduction, the authors should compare newer studies (e.g., 2020, 2021) about node classification tasks, and illustrate their strengths and weaknesses compared to this paper.

2.      This paper was compared with five baselines in experiments. However, the author should further compare with newer methods (e.g., last 3 years), which can show the advancement of proposed method.

3.      The paper sets the maximum training epochs as 200, so what is the criterion for judging the completion of model training? Do the similar models (e.g., deep learning model) adopt fair and uniform judgment criteria?

Author Response

Please see attachment below

Reviewer 4 Report

Dear Editor/Authors,

In the manuscript, "Scalably Using Node Attributes and Graph Structure for Node Classification" authors present the results on the task of node classification concern a network where nodes are associated with labels, but labels are known only for some of the nodes.

The topic is a kind of interest in different disciplines and methodologies. In the Introduction part, motivation and methods are well described and pointed out. The obtained results may be used for algorithms and studies’ improvement.

However, I had a look on the previous paper, namely the chapter in the book https://link.springer.com/content/pdf/10.1007/978-3-030-93413-2.pdf for which the current manuscript is considered as “an extended version of the paper”.   After comparing along with verifying the novelty of results in the manuscript with the paper I came to the following conclusions. The main algorithm and link to the code were developed in the previous paper. However, in a given current manuscript the same methods are described except a Jane-U(Batching) algorithm2 is presented on page 7, and results are compared with others in Tables, and figures. Some figures are taken from the previous paper without citing it ( e.g. fig3, partially others).

The paper is an extended version of the previous one but no novelty results or methods are considered here. From my point of view, the paper may be submitted as a review paper after the extended introduction and motivation part, but not as an original article taking into account the article https://link.springer.com/chapter/10.1007/978-3-030-93413-2_43

I would like to suggest the authors rewrite the manuscript because it cannot be published with existent content. 

Author Response

Please see attachment below

Round 2

Reviewer 3 Report

The author has made modifications to the manuscript in response to my questions and explained the reasons for the modifications in the reply letter.Therefore, the manuscript is already acceptable.

Reviewer 4 Report

Dear Authors and Editor,

I have no additional corrections or suggestions. And as far as the manuscript has been already discussed with Editor and accepted for further work I do not have objections as well.

With best wishes to you all